# Immunoadsorption for Treatment of Patients with Suspected Alzheimer Dementia and Agonistic Autoantibodies against Alpha1a-Adrenoceptor—Rationale and Design of the IMAD Pilot Study

**DOI:** 10.3390/jcm9061919

**Published:** 2020-06-19

**Authors:** Sylvia Stracke, Sandra Lange, Sarah Bornmann, Holger Kock, Lara Schulze, Johanna Klinger-König, Susanne Böhm, Antje Vogelgesang, Felix von Podewils, Agnes Föel, Stefan Gross, Katrin Wenzel, Gerd Wallukat, Harald Prüss, Alexander Dressel, Rudolf Kunze, Hans J. Grabe, Sönke Langner, Marcus Dörr

**Affiliations:** 1Department for Internal Medicine A, Nephrology, University Medicine Greifswald, Ferdinand-Sauerbruch-Straße, 17475 Greifswald, Germany; 2Institute of Diagnostic Radiology and Neuroradiology, University Medicine Greifswald, 17475 Greifswald, Germany; sandra.lange@uni-greifswald.de (S.L.); soenke.langner@med.uni-rostock.de (S.L.); 3Department of Neurology, University Medicine Greifswald, 17475 Greifswald, Germany; bornmanns@uni-greifswald.de (S.B.); antje.vogelgesang@uni-greifswald.de (A.V.); Felix.vonPodewils@med.uni-greifswald.de (F.v.P.); agnes.floeel@med.uni-greifswald.de (A.F.); 4Strategic Research Management, University Medicine Greifswald, 17475 Greifswald, Germany; holger.kock@uni-greifswald.de; 5Department of Psychiatry and Psychotherapy, University Medicine Greifswald, 17475 Greifswald, Germany; lara.schulze@uni-greifswald.de (L.S.); Johanna.Klinger-Koenig@med.uni-greifswald.de (J.K.-K.); Hans.Grabe@med.uni-greifswald.de (H.J.G.); 6Coordinating Centre for Clinical Trials, University Medicine Greifswald, 17475 Greifswald, Germany; Susanne.Boehm@med.uni-greifswald.de; 7German Center for Neurodegenerative Diseases (DZNE), 17475 Rostock/Greifswald, partner site Greifswald, Germany; 8Department of Internal Medicine B, University Medicine Greifswald, Ferdinand-Sauerbruch-Straße, 17475 Greifswald, Germany; stefan.gross1@uni-greifswald.de; 9German Centre for Cardiovascular Research (DZHK), 17475 Greifswald, Germany; 10Berlin Cures GmbH, 13125 Berlin, Germany; wenzel@berlincures.de (K.W.); wallukat@berlincures.de (G.W.); 11German Center for Neurodegenerative Diseases (DZNE) Berlin, 10117 Berlin, Germany; harald.pruess@charite.de; 12Department of Neurology and Experimental Neurology, Charité—Universitätsmedizin Berlin, 10117 Berlin, Germany; 13Department of Neurology, Carl-Thiem-Klinikum, 03048 Cottbus, Germany; a.dressel@ctk.de; 14Science Office, Hessenhagen 2, 17268 Flieth-Stegelitz, Germany; Rudolf.Kunze@gmx.de; 15Institute of Diagnostic and Interventional Radiology, University Medicine Rostock, 18057 Rostock, Germany

**Keywords:** Alzheimer’s clinical syndrome, dementia, immunoadsorption, autoantibodies, α1-Adrenergic receptor

## Abstract

Background: agonistic autoantibodies (agAABs) against G protein-coupled receptors (GPCR) have been linked to cardiovascular disease. In dementia patients, GPCR-agAABs against the α1- and ß2-adrenoceptors (α1AR- and ß2AR) were found at a prevalence of 50%. Elimination of agAABs by immunoadsorption (IA) was successfully applied in cardiovascular disease. The IMAD trial (Efficacy of immunoadsorption for treatment of persons with Alzheimer dementia and agonistic autoantibodies against alpha1A-adrenoceptor) investigates whether the removal of α1AR-AABs by a 5-day IA procedure has a positive effect (improvement or non-deterioration) on changes of hemodynamic, cognitive, vascular and metabolic parameters in patients with suspected Alzheimer’s clinical syndrome within a one-year follow-up period. Methods: the IMAD trial is designed as an exploratory monocentric interventional trial corresponding to a proof-of-concept phase-IIa study. If cognition capacity of eligible patients scores 19–26 in the Mini Mental State Examination (MMSE), patients are tested for the presence of agAABs by an enzyme-linked immunosorbent assay (ELISA)-based method, followed by a bioassay-based confirmation test, further screening and treatment with IA and intravenous immunoglobulin G (IgG) replacement. We aim to include 15 patients with IA/IgG and to complete follow-up data from at least 12 patients. The primary outcome parameter of the study is uncorrected mean cerebral perfusion measured in mL/min/100 gr of brain tissue determined by magnetic resonance imaging with arterial spin labeling after 12 months. Conclusion: IMAD is an important pilot study that will analyze whether the removal of α1AR-agAABs by immunoadsorption in α1AR-agAAB-positive patients with suspected Alzheimer’s clinical syndrome may slow the progression of dementia and/or may improve vascular functional parameters.

## 1. Introduction

Nearly 50 million people worldwide suffer from Alzheimer’s disease (AD) or other forms of dementia, and around 10 million new cases emerge every year, leading to a number of 150 million affected people expected in 2050 [1,2,3] Dementia has a lifetime prevalence ranging between 5% and 7% for those aged ≥60 years and is a major cause of disability among older adults [4,5] AD is the leading cause of dementia, responsible for two-thirds of all cases [6]. Since no causal treatment for AD is available yet, prevention strategies, psychosocial interventions and symptomatic pharmacological interventions are recommended and are central components of the treatment [7].

Up to now, research of causal therapies is focusing on the knowledge of typical neuropathological features of AD like amyloid plaques and neurofibrillary tangles which are associated with the tau-pathology [8,9]. In particular, the ß-amyloid hypothesis of AD has stimulated the development of therapy concepts directed against the amyloid protein and amyloid deposits in the brain of patients with AD.

One strategy is passive immunization with monoclonal antibodies which bind to ß-amyloid. Although it has been demonstrated that these antibodies may reduce the amyloid burden in the brain of AD patients, positive clinical effects were minimal or absent so far. Many promising compounds like Bapineuzumab, Gantenerumab or Solanezumab have failed in phase III of clinical trials or are still being evaluated (Aducanumab) [10,11,12].

Other ß-amyloid (Aß)-directed therapies focus on the enzymatic cleavage of the amyloid precursor protein (APP). It is known that ß-secretases contribute essentially to the production of Aß40/42 which is the toxic aggregating form of amyloid. Thus, ß-secretase inhibitors have been identified to be therapeutically beneficial. However, recently, a world-wide clinical trial on the secretase inhibitor Verubecestat was withdrawn because Verubecestat did not improve clinical ratings of dementia among patients with prodromal Alzheimer’s disease. Some measures even suggested an impairment of cognition and daily function compared to placebo [13].

The focus on Aß also led to the concept of removing it from plasma by therapeutic plasma exchange (TPE). Aß is bound to serum albumin by >90% which in turn is removed and discarded by TPE [14,15]. TPE-treatment with albumin replacement favored the stabilization of cerebral perfusion in mild to moderate AD patients compared to non-treated controls [15]. The same Spanish group currently conducts a prospective multicenter, randomized, blinded and placebo-controlled, parallel-group, phase IIb/III trial in patients with mild to moderate AD (“Alzheimer’s Management by Albumin Replacement (AMBAR)”). This study evaluates TPE with different replacement volumes of therapeutic albumin (5% and 20%), with or without intravenous immunoglobulins and is still ongoing [14]. Another group sought to remove Aß and developed an ex vivo adsorptive filtration system that resulted in an 80–100% reduction of Aβs within 30 min of circulation but has not yet been tested in humans [16].

In view of the numerous negative results, it seems to be necessary to shift attention to new therapeutic targets. In this respect, different pathologies of cognitive decline besides the Aß and the tau-pathologies may be considered. Importantly, clinical, pathological and epidemiological data point to a relevant overlap between cerebrovascular disease (CVD) and Alzheimer’s clinical syndrome [17]. Furthermore, cerebral microvascular lesions that are detected as white matter hyperintensities (WMH) on magnetic resonance imaging (MRI) are associated with typical gray matter atrophy patterns of AD in a considerable number of patients [18]. Thus, factors that alter the microenvironment of the endothelium and the smooth muscle cells of blood vessels may compromise the molecular exchange between blood and brain.

### Rationale of the Clinical Investigation

Naturally occurring agonistic autoantibodies (agAABs) against G protein-coupled receptors (GPCR) have been linked to cardiovascular disease such as dilated cardiomyopathy, myocarditis, malignant hypertension, vascular renal rejection, diabetes mellitus type 2 or dementia [19]. Agonistic AABs are functional antibodies that can activate the respective receptor. Agonistic AABs differ clearly from non-functional AABs. The latter trigger autoimmune disease in an Fc-receptor mediated manner whereas functional agAABs are able to bind cell receptors and activate intracellular signaling pathways that are normally triggered by endogenous ligands [20]. A pathological example of agAABs is Graves’ hyperthyroidism with autoantibodies activating the thyroid-stimulating hormone (TSH)-receptor and with subsequent overproduction of thyroid hormones [21]. Other examples for GPCR-agAABs are AABs directed against adrenoceptors (AR; e.g., ß1AR and ß2AR, α1AR), the angiotensin (2) receptor type 1 (AT-R1) and the endothelin receptor type A (ETA) [19]. The agAABs against AR are directed against the first or second extracellular loop of the receptor. They bind to constant epitopes defined by the amino acid sequence in the respective loop. These AABs belong to the immunoglobulin class G (isotypes 1–3). GPCR-agAABs elicit a long-lasting dimerization of adrenoceptors and continuously activate cellular processes such as phosphorylation of intracellular proteins and modulation of calcium signaling that results e.g., in the case of agAABs against the α1AR in the proliferation of smooth muscle cells and thickening of vessel walls [22]. In response to their natural agonists, the receptor density on the cell membrane is regulated by receptor desensitization. This mechanism is inhibited by agAABs [22]. In an animal model, continuous stimulation triggered by α1AR-AABs leads to cerebrovascular remodeling and obliteration [23].

In a small clinical trial, treatment of Graves’ disease with rituximab, a B-cell depleting monoclonal antibody specifically reduced the production of TSH-receptor antibodies [24]. In a case of thyreotoxic crisis, TPE is also used to remove agAABs against TSH-receptor and albumin-bound thyroid hormones [25].

Cerebrovascular remodeling may lead to disturbances in cerebral blood flow and a lack of outflow of Aß. A clearing defect for Aß rather than an Aß-overproduction has been proven experimentally for AD patients [26]. From this view, AABs against the α1AR may interfere with Aß clearance mechanisms and act as a risk factor and a modulating component of dementia in patients with Alzheimer’s clinical syndrome.

In line with this, the GPCR-α1AR was also reported to be a target for agAABs in patients with essential and malignant hypertension [27]. In a pilot study in five α1AR-AAB-positive patients with resistant hypertension, removal of these AABs by immunoadsorption lowered the mean arterial blood pressure significantly by approximately 10 mmHg, and the effect was still present after 180 days [28]. In line with this, a recent study in 816 subjects showed that the occurrence of α1AR-AABs predicted arterial stiffness progression even in normotensives over a 5-year period [29].

A disease that has been very well investigated with regard to agAABs is dilated cardiomyopathy (DCM). Among others, agAABs against the GPCR-ß1-adrenoceptor (ß1R) seem to play an important role in DCM. Their elimination from the blood by immunoadsorption (IA) has been transferred to a therapeutic intervention in clinical praxis. Removal of circulating antibodies by IA with subsequent intravenous immunoglobulin G substitution (IA/IgG) has been shown to result in improvement of cardiac function, in better exercise capacity, and in decrease of myocardial inflammation in DCM [30,31,32,33]. Removal of functional AABs does not have to be a constant process as in non-functional AAB triggered disease. Already a one-week-course of IA seems to yield a so-called legacy effect that may persist for a long time [34]. In DCM, removal of the agAABs by a one-week-course of IA lasts for up to 12 months and even longer [35,36].

Noticeably, in dementia patients, agAABs against both the α1R- and ß2R-adrenoceptors were also found at a prevalence of approx. 50%. Thus, in a primary care cohort who screened positive for dementia, 40 out of 95 participants were also positive for agAAB (29 subjects with α1AR-AABs and 21 with ß2AR-AABs) [37]. However, agAABs could not discriminate between Alzheimer’s Dementia and other forms of dementia. Patients with coronary heart disease were more likely (OR = 4.23) to have α1AR-AABs than those without coronary heart disease. The presence of agAAB against adrenoceptors, especially α1AR-agAABs, in persons suffering from dementia motivated a first pilot trial on the effects of IA on the course of dementia [38]. In this trial, in four out of eight patients an effective depletion of agAAB could be achieved by a 4-day per-protocol IA treatment, while another four patients received a less effective 2–3 day treatment due venous access problems. IA was safe to use in these patients, and the mean change in Mini Mental State Examination (MMSE) score of these patients remained constant over 12 to 18 months.

The IMAD trial (Efficacy of immunoadsorption for treatment of persons with Alzheimer dementia and agonistic autoantibodies against alpha1A-adrenoceptor) aims to ascertain whether the positive effects of IA on slowing down dementia progression can be replicated. Moreover, IMAD will comprehensively examine potential effects of this treatment in patients with Alzheimer’s clinical syndrome by a combination of brain and vessel imaging along with cognitive tests and further cardiovascular, cerebrovascular and laboratory examinations.

## 2. Methods

### 2.1. Objectives

The IMAD trial is designed as an exploratory monocentric interventional trial corresponding to a proof-of-concept phase-IIa study. The aim of IMAD is to evaluate whether IA with subsequent IgG-substitution (IA/IgG) is related to an improved uncorrected mean brain perfusion after 12 months as a surrogate for potential beneficial effects on disease progression in patients with an Alzheimer’s clinical syndrome and mild to moderate cognitive impairment. In addition, potential effects of IA/IgG on cognitive measures as well as cardiovascular, cerebrovascular and laboratory parameters will be investigated.

### 2.2. Patients

Potential participants for this trial need to have an Alzheimer’s clinical syndrome, according to the definition as outlined in Jack et al. [39]. In patients without obvious exclusion criteria, cognitive capacity is again assessed before inclusion using the MMSE. If a mild to moderate impairment (defined by a MMSE score between 19 and 26) is confirmed, patients are tested for the presence of agAABs by an ELISA-based method, followed by a bioassay-based confirmation test (methodological details are given in Section 2.5.8). Patients with a positive test result by the bioassay are eligible for further screening. We aim to include in total 15 patients with IA/IgG and 12 patients with complete follow-up data.

### 2.3. Inclusion and Exclusion Criteria

Criteria that have to be fulfilled for all participants (inclusion criteria) and reasons that prevent inclusion into the study (exclusion criteria) are summarized in Table 1.

### 2.4. Study Design

During the screening phase, patients are checked for eligibility according the inclusion and exclusion criteria (Table 1). Patients fulfilling these criteria are comprehensively examined during a baseline visit, followed by IA/IgG treatment. All participants of the IMAD trial are followed up after 1, 6 and 12 months. The complete study flow is illustrated in Figure 1.

The IMAD trial is an interdisciplinary project which involves the departments and institutes of radiology, neurology, psychiatry, cardiology, nephrology and laboratory medicine of the University Medicine Greifswald. The study complies with the Declaration of Helsinki and it has been approved by the Ethics Committee of the University of Greifswald (MPG 02/16; MPG 02/16a). The trial is registered at ClinicalTrials.gov (NCT03132272).

### 2.5. Examinations and Assessments

Patients undergo a comprehensive examination program including brain perfusion assessment by MRI (primary outcome parameter: uncorrected mean brain perfusion assessed by arterial spin labeling [ASL]) and assessment of further structural and vascular brain MRI parameters. In addition, several cognitive, cardiovascular, cerebrovascular and laboratory parameters are assessed during baseline and all follow-up visits. Table 2 gives an overview about methods, main parameters and respective time points.

For examinations with a high potential for an observer bias (e.g., MRI, echocardiography), images will be stored and the reading will be done offline in a blinded manner. Specifically, they will be assessed without knowledge of the respective patient and time point. All examinations and methods used are described in more detail in the following sections.

#### 2.5.1. Brain Magnetic Resonance Imaging (MRI)

##### MR Imaging Protocol

All subjects undergo brain imaging at 1.5T (Magnetom Aera, Semens, Germany) using a 20-channel head coil for image acquisition. Structural MR imaging protocol includes a sagittal 3D T1-weighted sequence with an inplane resolution of 1 × 1 mm and a slice thickness of 1.3 mm (repetition time [TR] = 1860 ms, echo time [TE] = 3.88 ms, inversion time [TI] = 1000 ms, 160 slices); a sagittal 3D FLAIR dataset with 1 × 1 mm inplane spatial resolution and 1.1 mm slice thickness (TR = 5000 ms, TE = 214 ms, TI = 1800 ms, field of view [FoV] = 265 × 265 mm); a diffusion weighted imaging (DWI) sequence with b-values 0/1000 s/mm^2^ (TR = 5600 ms, TE = 113 ms, 1.2 × 1.2 mm voxel size, 5 mm slice thickness) and a time-of-flight angiography of the circle of Willis with 0.5 × 0.5 mm spatial resolution with a slice thickness of 0.8 mm (TR = 31 ms, TE = 7.15 ms, FoV = 200 mm).

Cerebral perfusion is assessed using a 2D pseudo-continuous arterial spin-labeling sequence (PICORE Q2T) with 5 mm slice thickness and an in-plane spatial resolution of 4 mm with 64 slices. Other imaging parameters are a post labeling delay of 1.8 s, bolus duration of 700 ms, TR = 2500 ms and TE = 13 ms.

A diffusion tensor imaging (DTI) dataset is acquired in all patients (TR = 4700 ms, TE = 116 ms, b-value 0/1000 s/mm^2^, 12 directions) with a slice thickness of 4 mm and an in-plane spatial resolution of 1.5 × 1.5 mm.

Total acquisition time is 42 min. All scans are checked by a board certified neuroradiologist for gross abnormalities.

##### MR Image Analysis

For structural image analysis, all MR datasets are transferred to a dedicated Horos workstation (www.horosproject.org). WMH are evaluated on axial reconstructions of the 3D FLAIR dataset with 3 mm slice thickness according to the Fazekas scale [40]. The Fazekas score are dichotomized into low (Fazekas grade 0–1) and high (Fazekas grade 2–3). Cerebral microbleeds are defined as hypointense lesions smaller than 10 mm on b = 0 diffusion weighted images. Lacunar lesions are defined as small lesions in the deep white and grey matter with a diameter between 3 and 10 mm and cerebrospinal fluid-like signal on all sequences. For quantitative analysis, lacunar lesions and microbleeds are counted by visual inspection and for further statistical analysis dichotomized in present or absent.

Medial temporal lobe atrophy (MTA) score is rated on coronal reformations of the 3D T1w dataset according to previously described criteria [41]. For statistical analysis, the mean MTA score of both sides is dichotomized into high (>1.5) and low (≤1.5) as described elsewhere [42].

Vessel diameter of the internal carotid artery (ICA), the anterior (ACA) and middle cerebral artery (MCA) and the basilar artery (BA) are evaluated by manual measurements. For the ICA, measurements are performed at the level of the cavernous sinus, for the MCA in the median M1 segment, for the ACA immediately distal the anterior communicating artery and for the basilar artery at the level of the origin of the superior cerebellar artery, respectively.

Brain volume estimation are performed using T1w images. Therefore, the measured raw DICOM data are converted into NIFTI (Neuroimaging Informatics Technology Initiative) format using dcm2nii, which is part of the neuroimaging tool MRIcron. Preprocessing using FSL (version 6.0, www.fsl.fmrib.ox.ac.uk/fsl) included correction for gradient nonlinearities, non-brain tissue removal, linear registration to standard space, and tissue segmentation [43]. Evaluation of the pseudo-continuous ASL images is performed as previously described by Binnewijzend et al. [43]. Therefore, ASL images are also corrected for gradient nonlinearities in all three directions and then linearly registered to the brain-extracted T1-weighted images. The brain mask is used to calculate uncorrected mean whole-brain cerebral blood flow (CBF). These volume estimates are then transformed to the ASL data space to correct partial volume-corrected cortical and white matter CBF maps [44]. CBF values are also extracted using regions of interest (ROIs) in the frontal, temporal, occipital and parietal brain areas and the hippocampus based on the MNI152 atlas and the Harvard–Oxford cortical atlas. The primary outcome parameter of the study is uncorrected mean cerebral perfusion measured in mL/min/100 gr of brain tissue determined by ASL.

Preprocessing of DTI data includes also conversion to NIFTI format. Then, the tool eddy correct, part of FSL, is used to correct the diffusion-weighted data with respect to subject motion and deformations introduced by eddy current artifacts of the MRI scanner. Fractional anisotropy (FA) images are created by fitting a tensor model to the raw diffusion data using FSL DTI-FIT. FA analyses was performed on a whole brain basis and using the ROI from CBF-analyses.

#### 2.5.2. Cognitive Assessment

##### Mini-Mental State Examination (MMSE)

We use MMSE-2 which is a revised version of the original MMSE [45,46] routinely used to measure cognitive decline. The MMSE-2 consists of three versions: the standard version, the brief version and the expanded version of the MMSE. In this study, the standard version is used in order to maximize the benefit of the use of the scale while minimizing the duration of the cognitive assessment. The MMSE-2 shows a sufficient internal consistency (Cronbach’s alpha 0.66–0.79) [46].

##### Alzheimer’s Disease Assessment Scale - Cognition (ADAS-Cog)

The Alzheimer’s Disease Assessment Scale (ADAS) is commonly used to assess cognitive dysfunction in individuals with Alzheimer’s Disease and other types of dementia [47,48]. The ADAS-Cog was developed as a two-part scale: one that measures cognitive impairment and one that measures non-cognitive factors such as mood and behavior. In IMAD, only the cognitive scale of the ADAS-Cog is applied, which consists of 11 parts and measures the cognitive functioning, language, and memory in a 30-min test. Five parallel versions are available to avoid recall bias due to multiple testing. The final score ranges from 0 to 70 points, with higher scores indicating more serious cognitive impairment. The ADAS shows a good internal consistency (Cronbach’s alpha 0.61–0.76) [49]. During the baseline visit, a cognitive profile over the 11 dimensions measured by the ADAS-Cog is created. The profile is compared to reference values provided by Graham et al. [48] (Figure 2). According to the reference values, the cognitive profile of patients with mild to moderate AD predominately shows higher impairment in memory, and to a lesser extent, in cognitive functioning. In contrast, language is hardly impaired in this stadium (Figure 2). Thus, patients with a different cognitive profile undergo further examinations to exclude differential diagnoses. For patients included into the IMAD trial, a cognitive change is estimated according to Stern et al. [50] to predict future cognitive decline during the follow-up period. The estimated cognitive change is later compared to the observed cognitive changes over 12 months after IA/IgG.

##### Verbal Learning and Memory Test (VLMT)

The Verbal Learning and Memory Test (VLMT) allows a short and individual assessment of verbal learning and memory. The VLMT [51] uses 15 semantically independent words to assess verbal memory. In a second step, an interference list with 15 new words is learned and recalled to distract from the first, target list. After 20–30 min a delayed recall of the target list and a recognition test with a new list combining the 30 learned words with 20 semantically and phonemically similar words is done.

##### Benton Test

The Benton Test (first edition: Benton, 1946) is a visual retention test for clinical use testing the memory of visuo-spatial stimuli. The patient has to reproduce, draw or recognize presented graphic stimuli. In this trial, instruction A (10 figures shown for 10 s) and parallel forms C, D and E are used. The Benton Test has been shown to have a high internal consistency (Cronbach’s alpha = 0.94–0.98) and validity [52].

##### Geriatric Depression Scale (GDS)

The Geriatric Depression Scale (GDS) is used to measure depressive symptoms in older persons. It is a 15-item questionnaire demanding a dichotomous (yes/no) evaluation of depressive symptoms. It was also shown to be applicable for persons with advanced cognitive impairment [53]. The GDS shows a high internal consistency (Cronbach’s alpha 0.91).

The standardized questionnaires used in the IMAD trial as well as the respective cut-off values applied are summarized in Table 3.

#### 2.5.3. Cardiovascular Examinations

All cardiovascular examinations are conducted according to standardized procedures by certified study nurses and physicians at the cardiovascular examination center of the German Centre for Cardiovascular Disease (DZHK) in Greifswald. The following examinations are part of the IMAD phenotyping:

##### Brachial Blood Pressure Measurement

Measurements of the brachial blood pressure are taken using an Omron 705 IT (OMRON Healthcare Europe) blood pressure monitor with appropriate cuff size after a resting period of at least 5 min in a sitting position. Accordingly, three measurements are taken on the right arm, with a 3-min break between each measurement [54]. All individual measurements (systolic blood pressure [mmHg], diastolic blood pressure [mmHg], heart rate [/min]) are recorded.

##### Pulse-Wave Analysis and Central Hemodynamics

To perform cuff-based non-invasive data capturing at the brachial artery, the invasively validated Mobil–O–Graph pulse-wave analysis (PWA) monitor (IEM GmbH, Stolberg, Germany) with inbuilt ARCSolver algorithm is used [55,56]. After conventional blood pressure measurements, the brachial cuff is inflated additionally to the diastolic blood pressure level and held for about 10 s to record pulse waves. Subsequently, central pressure curves are automatically obtained through a transfer function. In total, three measurements are taken, with a 3-min break between each measurement. The following parameters are assessed by this method: pulse-wave velocity (PWV [m/s]), augmentation index (Aix [%]), heart-rate corrected augmentation index (Alx@75 [%]), central systolic blood pressure (cSBP [mmHg]), and central diastolic (cDBP [mmHg]).

##### Digital Endothelial Vascular Function and Stiffness

Digital pulse amplitude is measured with a pulse amplitude tonometry device placed on the tip of the right index finger (Endo-PAT2000, Itamar Medical, Caesarea, Israel) [57]. This device comprises a pneumatic plethysmograph that applies uniform pressure to the surface of the distal finger, allowing measurement of pulse volume changes. Throughout the study, the inflation pressure of the digital device is electronically set to 10 mm Hg below diastolic blood pressure or 70 mm Hg (whichever is lower). Baseline pulse amplitude is measured for 2 min 20 s. Arterial flow is interrupted for 5 min by a cuff placed on a proximal forearm using an occlusion pressure of 200 mm Hg or 60 mm Hg above systolic blood pressure (whichever is higher). The pulse amplitude is recorded electronically and analyzed by the computerized, automated algorithm of the device that provides the average pulse amplitude for each 30-s interval after forearm cuff deflation up to 4 min [57]. The following parameters which are known markers of endothelial function and vascular stiffness is derived from these measurements: augmentation index (Aix [%]), heart-rate corrected augmentation index (AIx@75% [%]), reactive hyperemia index (RHI) [57].

##### Transcutaneous Oxygen Pressure (tcPO2)

Transcutaneous oxygen pressure (tcPO2) measurements are performed with the PRÉCISE 8008 device (medicap GmbH, Ulrichstein, Germany). After a resting period of at least 10 min in supine position, four probes are placed at the dorsum of the feet and at the back of both hands. Measurements are taken while patients are breathing ambient air, in a resting supine position at room temperature, between 22 °C and 25 °C. The site on the foot is carefully cleaned before the probes are applied to the skin, using adhesive rings and contact liquid, supplied by the manufacturer. The measurements are performed after calibration and preheating of the transducer to approximate 44 °C [58]. After termination of the procedure tcPO2 [mmHg] values for the four measurement sites are recorded.

##### Echocardiography

Transthoracic echocardiography as a non-invasive gold standard for the determination cardiac function and morphology is performed by certified physicians (Vingmed Vivid 9, 5S transducer 2.0–5.0 MHz, GE Medical Systems GmbH, Hamburg, Germany). All images and loops are stored digitally and are analyzed offline. The reading of the echocardiograms is performed according to current recommendations [59] includes parameters of left atrial and left ventricular (LV) structure (left atrial diameter in parasternal short axis [mm]; left atrial volume in 4 chamber view [cm^2^], enddiastolic/endsystolic thickness of the intraventricular septum and posterial wall [mm], LV mass [g], enddiastolic/endsystolic LV volume [mL]) as well as LV systolic and diastolic function (biplane LV ejection fraction according to Simpsons rule [%], global longitudinal strain [%], peak velocity of the mitral E- and A-wave [cm/s], deceleration time of the mitral E-wave [ms], isovolumetric relaxation time [ms], peak velocity of the excursion of the lateral and septal mitral annulus in the early diastolic phase [cm/s], ratio between the peak velocity of the excursion of the mean lateral/septal mitral annulus in the early diastolic phase and the peak velocity of the mitral E-wave).

#### 2.5.4. Kidney Function and Ultrasound

Kidney function is determined by blood and urinary laboratory tests: estimated glomerular filtration rate by serum creatine and urinary albumin–creatinine ratio.

Renal ultrasound is performed with a HITACHI EUB-7500 machine. Kidney length is determined as the maximum longitudinal dimension. Parenchymal thickness is measured as the shortest distance from the renal sinus fat to the renal capsule at three different points: at the upper and lower pole and at the middle. The parenchymal-pyelon-index is calculated as the sum of ventral and dorsal parenchymal thickness (in a cross-section of the kidney) divided by the width of the central echo complex. The following categories are generally assessed: location, anomalies as agenesis, hypo- or hyperplasia, horseshoe kidney; kidney length; kidney width; parenchymal thickness; surface roughness; echogenicity; parenchymal-pyelon-index; medullary or parenchymal calcification; number and size of cysts, stones, infraction zones and tumors [60].

#### 2.5.5. Ultrasound of the Extracranial Arteries

Ultrasound of the extracranial arteries is performed with a Philips UI 22 machine. Extracranial carotid and vertebral arteries (VA) are examined with linear ultrasound transducers (bandwidth 3–13 MHz). Systolic, diastolic, and mean flow velocities in common carotid artery, internal carotid artery (ICA), and V2 segments of VA are documented after angle correction. We classify ICA stenosis uniformly according to current ultrasound criteria for grading internal carotid artery stenoses of the the German Society of Ultrasound in Medicine (DEGUM) and Transfer to grading system of the North American Symptomatic Carotid Endarterectomy Trial (NASCET) [61]: if peak systolic velocity (PSV) is ≥125 cm/s, ICA stenosis is defined as being equivalent to ≥50% according to North American Symptomatic Carotid Endarterectomy Trial criteria. Occlusion is defined by absence of Doppler and color signal, typical proximal biphasic Doppler spectra, and additional indirect criteria like crossflow. Carotid plaque is defined as any arterial wall irregularity thicker than 1.5 mm or exceeding >50% of the surrounding wall thickness that protruded into the vessel lumen.

VA measurements are taken in the V2 segment and considered abnormal if there is direct evidence of local or indirect evidence of proximal or distal flow abnormalities. Overall, abnormal flow characteristic of the posterior circulation is defined by at least unilateral (1) flow abnormality of the (extracranial) V2 segment of either VA, (2) intracranial VA stenosis or occlusion, or (3) basilar artery (BA) stenosis or occlusion.

#### 2.5.6. Blood and Urine Samples

Blood and urine samples are obtained according to standard operating procedures. In total, 403.7 mL blood and 42.0 mL urine are obtained per participant (250.0 mL for serum analysis, 121.5 mL for plasma analytics, 16 mL EDTA, 16.2 mL Citrate). At the study center, samples are analyzed immediately after blood and urine sampling. Two aliquots of 0.5 mL serum are stored in a freezer (−80 °C) for further analysis.

For sample analyses at independent laboratories (see Section 2.5.8), blood samples (170.0 mL for serum analytics, 49.5 mL for plasma analytics) are collected at the study center. Further sample management is accomplished by biometec GmbH, Greifswald, Germany.

Blood samples are obtained according to standard operating procedures. In total, 29 mL were obtained per participant (8.5 mL for serum analysis, 8 mL for plasma analytics, 10 mL EDTA, 2.5 mL for blood RNA). At the study center they were stored in a freezer at −80 °C). For this analysis, 100 aliquots were available.

#### 2.5.7. Laboratory Parameters

A complete list of the laboratory parameters can be found in the Appendix A (Table A1). Laboratory analytics of blood and urine are carried out in accordance with established standard operating procedures and preanalytical protocols follow the schemes of the GANI_MED (Greifswald Approach to Individualized Medicine) project [62].

#### 2.5.8. Assessment of Autoantibodies

An enzyme-linked immunosorbent assay (ELISA) is used to detect agAAB as described previously [37,63]. Analyses are performed by an independent laboratory (E.R.D.E.-AAK-Diagnostik GmbH, Berlin, Germany) blinded to clinical patient data. In brief, peptides are directed against the ß1-adrenergic receptor loop 1 and ß2- adrenergic receptor loops 1 and 2. Modified peptides are bound to 96-well streptavidin-coated plates. Peptides are coupled to preblocked streptavidin-coated 96-well plates (Perbio Science, Bonn, Germany). Patient serum is added in a 1:100 dilution and incubated for 60 min. As detection antibody a horseradish peroxidase conjugated anti-human IgG antibody is used (Biomol, Hamburg, Germany). Antibody binding is visualized by the 1-Step Ultra TMB ELISA (Perbio Science, Bonn, Germany). The absorbance is measured at 450 nm against 650 nm with an SLT Spectra multiplate reader (TECAN, Crailsheim, Germany).

As confirmation test, a bioassay is used that has been established by Wallukat and Wollenberger for the identification and quantification of GPCR-AABs [64], and that has been modified and standardized as described previously [65,66]. Analyses are performed by an external laboratory (Berlin Cures GmbH, Berlin, Germany) without knowledge about any further patient characteristics or parameters. In this bioassay, the chronotropic response of spontaneously beating cultured neonatal rat cardiomyocytes to patients’ IgG-containing GPCR-AABs is recorded [67].

Anti-NMDA (N-methyl-D-aspartate) autoantibodies are determined in the participant’s sera by immunohistochemistry according to manufacturer’s instruction (Anti-Glutamate-Receptor-IgG (Typ NMDA)–IFFT, EUROIMMUN, Lübeck, Germany).

Additional measurements are performed by biometec GmbH, Greifswald. This includes analyzes of antibodies against oxidized low-density lipoprotein (oxLDL) and β-amyloid, vasculitis marker (aab against myeloperoxidase (anti-MPO), proteinase 3 (anti-PR3), glomerular basement membrane (anti-GBM)), B-cell activity and antibody development (B-cell activating factor (BAFF)) and neurodegeneration (neurogranin).

### 2.6. Intervention

Immunoadsorption is performed with ADAsorb apheresis devices equipped with Globaffin adsorber columns in the dialysis department of the University Medicine Greifswald.

The Globaffin column is a regenerative twin adsorber system that utilizes peptide ligands (Peptid-GAM^®^; Fresenius Medical Care, Bad Homburg, Germany) covalently bound to sepharose. These peptide ligands have a strong affinity for Fc fragments of immunoglobulins from any source and selectively remove immunoglobulins and immune complexes from plasma without affecting other plasma proteins. The Globaffin adsorber system has previously been used in antibody mediated disorders, such as dilated cardiomyopathy and acute renal transplant rejection [68].

In contrast to unselective plasma exchange where all plasma components including albumin, clotting factors and immunoglobulins are discarded and replaced with a fluid containing either albumin and colloid or donor plasma, immunoadsorption is a semi-selective device. First, the patient’s plasma is separated from blood cells by a membrane. Plasma is then passed over the Globaffin twin adsorber system which selectively remove IgG, IgA, and IgM before the plasma is re-infused to the patient.

Typically, approximately four column volumes (250 mL) are processed before the plasma stream is directed to the second column within the device and the first column undergoes a four-step regeneration, so that the apheresis cycle can be reiterated: (1) replacement of residual plasma by 0.9% NaCl solution; (2) elution of bound immunoglobulin by hydrochloric acid [glycine-HCl] buffer at pH 2.8; (3) neutralization by phosphate buffered saline [PBS] and 4. replacement of PBS by 0.9% NaCl (Figure 3).

A separate twin-column pair is assigned to each patient. A total of approximately 2.0-fold of blood plasma volume is processed per day (approximately 4 to 6 h per session) on five consecutive days. After the immunoadsorption series, the patients receive 500 mg per kg body weight intravenous immunoglobulin substitution on day 6 (during approximately 6 h). The dose is determined based on body weight and amounts to 500 mg/kg, depending on the packaging size. The eluates of the regeneration of the columns are collected automatically and are used for detailed immunological analyses. The patients are hospitalized for the immunoadsorption procedures and Ig substitution period in the neurological department of the University Medicine Greifswald.

### 2.7. Statistical Considerations

The objective of this trial is to assess the effects of IA/IgG in patients with agAABs and dementia in patients with suspected Alzheimer’s clinical syndrome. The primary target parameter is uncorrected mean cerebral brain perfusion as assessed by ASL 12 months after treatment.

#### 2.7.1. Statistical Analyses

Analyses will follow an exploratory approach since we plan n = 15 subjects but do not consider a comparison group. Although the main focus of analysis is descriptive, there is an interest in comparing the development of outcomes of 1 month before and 12 months after IA/IgG.

Effect sizes will be estimated using appropriate (generalized) linear mixed regression models. If necessary non-parametric models will be used. To evaluate effect sizes, suggestions according to Cohen (1988) are used [69]. Analyses are performed as intention-to-treat (ITT) and per-protocol (PP).

#### 2.7.2. Missing Data

Missing values are not replaced with substituted values. Due to the small sample size and the exploratory character of the investigation, imputation techniques are not recommended.

#### 2.7.3. Effect Size Consideration

Assuming a prevalence of agAABs of 31.5% and a drop-out rate of 20%, 120 participants have to be checked for eligibility to reach the aim of 15 participants enrolled in the intervention. With a sample size of n = 12, an expected α-error of 5% and a power of 80%, standardized effect sizes of 0.766 can be detected (for one-sided paired *t*-tests), thus allowing a possible drop out of maximal 3 participants.

For data analyses, Stata (Version 14, StataCorp, College Station, TX, USA), SPSS statistics version 22 (IBM Corp., Armonk, NY, USA) or MATLAB (Version R2015a, Mathworks Inc., Natick, MA, USA) will be used.

## 3. Discussion

The IMAD study investigates a new pathophysiological and therapeutic aspect of Alzheimer’s clinical syndrome, the removal of α1AR-agAABs by immunoadsorption in patients with cognitive impairment and suspected AD. Outcome parameters comprise cerebral blood flow measured by arterial spin labelling MRI (primary), cognition measured by validated cognitive tests and other questionnaires (ADAS-Cog, MMSE, VLMT, Benton Test, GDS) and vascular effects assessed by echocardiography, sonography, blood pressure, pulse wave velocity, plethysmography and transcutaneous oxygen measurement.

Extracorporeal therapies for dementia in Alzheimer’s clinical syndrome and CVD are innovative therapeutic options. Recently, three different medical devices have been tested: dialysis, TPE and IA. The Spanish AMBAR study is currently examining whether the peripheral lowering of Aß by TPE and concurrent albumin substitution has an impact on cognitive performance [70]. Kitaguchi and colleagues use dialysis systems [71] and adsorptive double-filtration systems [8] to lower the plasma levels of Aß. Both groups are assuming that the removal of Aß may reduce the cerebral Aß load. By using IA, we contrary aim to target vascular effects of α1AR-agAABs and only secondarily at a probably better clearance of Aß. The IMAD study can profit from the experience of a previously performed open pilot trial [38]. In this earlier study, the applicability of apheresis to dementia patients and safety aspects were examined. The sustainability of the elimination of α1AR-agAABs was proven and the first indications of a stabilization of cognitive performance were observed. The IMAD study now investigates whether the removal of α1AR-AABs by a 5-day IA procedure has a positive effect (improvement or non-deterioration) on impairment-relevant hemodynamic, cognitive, neurological, vascular and metabolic parameters within a one-year follow-up period.

As an exploratory trial, the IMAD study has, owing to feasibility constraints, a small projected sample size in a monocentric, single-arm and unblinded design. Thus, only large effects can reach statistical significance and, even then, the absence of a control group and other trial site(s) will still confine the validity of the results. Nevertheless, this trial may provide important insights whether eliminating or reducing α1AR-agAABs as a contributing factor of dementia-related cerebrovascular impairment opens up a completely new treatment approach for α1AR-agAABs-positive persons along the course of dementia progression in patients with Alzheimer’s clinical syndrome. It is of course possible that other agAABs also play a role in the disease course, e.g., ß2AR which has been found in dementia patients in previous studies [37,38].

In this respect, the comprehensive and extensive set of measured endpoints has the potential to indicate possible intervention effects on a broad (patho)physiological spectrum. Indeed, the trial protocol has been devised deliberately to comprise as many measurable physiological and metabolic parameters as possible besides the neurocognitive tests. Therefore, the IMAD trial results will allow to correlate intervention effects with potential physiological or functional mode(s) of action. Such correlations may form the basis of targeted, larger and statistically more robust trials to specifically and precisely uncover the effects of immunoadsorption on affected patients. In the future, the optimal intervention time during the disease progression and the determinants that predict and govern the response profile should be addressed in order to achieve a maximal beneficial effect of IA. In this regard, one challenge will be to pinpoint the actual pathomechanistically active autoantibodies or autoantibodies to delineate less-invasive specific depletion or inactivation schemes.

However, if the IA treatment approach does turn out to have a beneficial capacity for at least a well-defined subgroup of patients at risk of dementia progression in Alzheimer’s clinical syndrome, its one-time character (a single week of hospitalization) will certainly be advantageous, discarding or at least attenuating the pharmacotherapeutic need for long-term compliance adherence. For reasons that have not yet been clarified, the GPCR-AABs seldom reoccur after being removed—both in DCM [36,72,73] and in dementia [38].

In the case of a positive outcome of the planned study, functional vascular improvement and cognitive stability or improvement over at least 12 months, knowledge and experience should have been gained to start a well-planned controlled, prospective, multicenter and randomized clinical study. Its data could then be used to prove that IA is suitable for the treatment of mild and moderate dementia with vascular pathological AABs and complements antidementive drug therapies with other targets.

Our study design has potential strengths and limitations that merit further discussion. We see the interdisciplinary approach where knowledge from different disciplines and viewpoints are combined as a unique strength. A further strength resulting directly from this is the comprehensive phenotyping with different end-points to generate a broad spectrum of results that may help design a subsequent multicenter pilot study. As a potential limitation we see the small sample size with low statistical power. For this reason, we are also not able to investigate other potential risk factors (e.g., genetic disposition). As we include patients with cognitive impairment without additional testing of CSF biomarkers we face limitations in the diagnostic classification of their syndromes. Thus, we prefer to label their impairments as suspected or probable AD. On the other hand, we use well-validated neuropsychological testing (MMSE, ADAS-cog, Benton Test, VLMT) and carefully exclude many medical conditions that could lead to secondary and potentially treatable dementia. Moreover, based on the MRI scans we can exclude patients with organic/structural brain damages from the study. In fact, our intention behind this patient selection was the inclusion of patients in their beginning or early phase of probable AD to detect possible changes due to immunoadsorption in physiological and cognitive parameters which might not occur in later states of dementia and to ensure the ability of collaboration and adherence throughout a 12-months follow-up period of the study. Furthermore, we may miss results in the long-term for future associations and detection of causalities due to the relatively short follow up time of 12 months. Additionally, knowledge about the prevalence of agonistic autoantibodies in other forms of dementia and in the general population is limited. Keeping these limitations in mind, our study is designed as an exploratory study and aims at showing proof of principle.

## 4. Conclusions

IMAD is an important pilot study that will analyze whether the removal of α1AR-agAABs by immunoadsorption in α1AR-agAAB-positive persons slows the progression of dementia in Alzheimer’s clinical syndrome and/or improves vascular functional parameters.

## Figures and Tables

**Figure 1 jcm-09-01919-f001:**
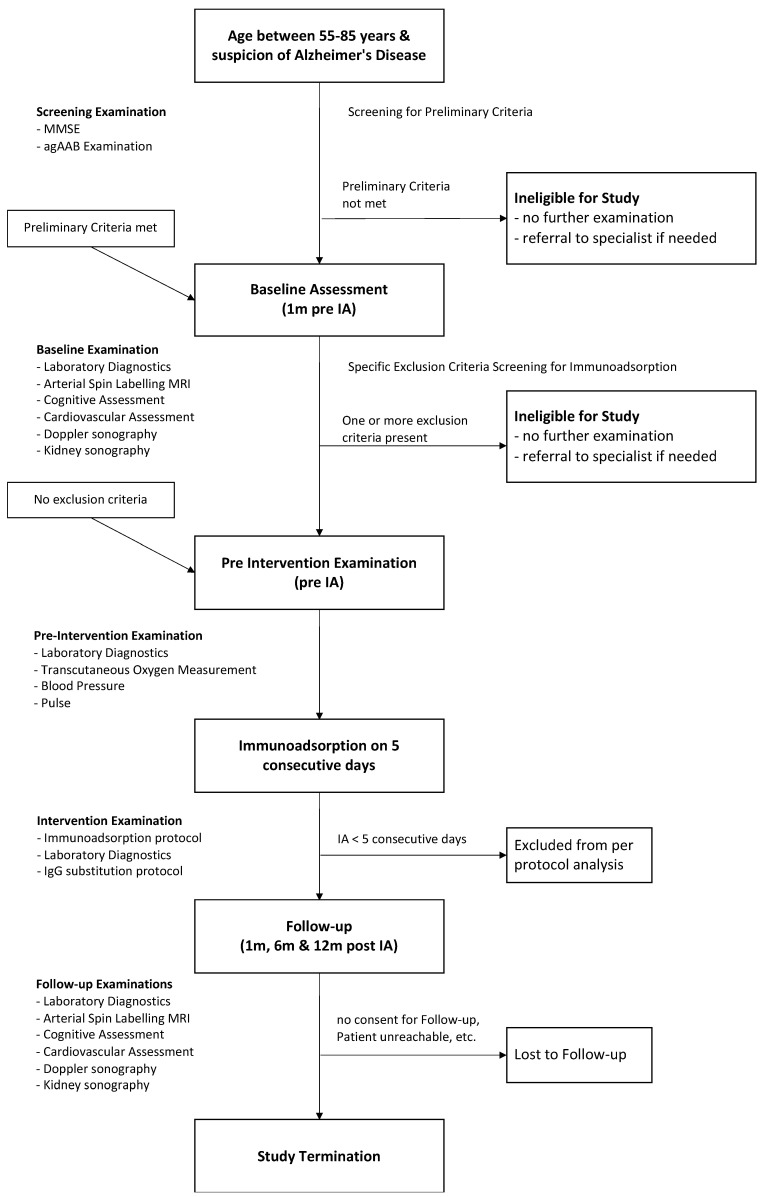
Study flow.

**Figure 2 jcm-09-01919-f002:**
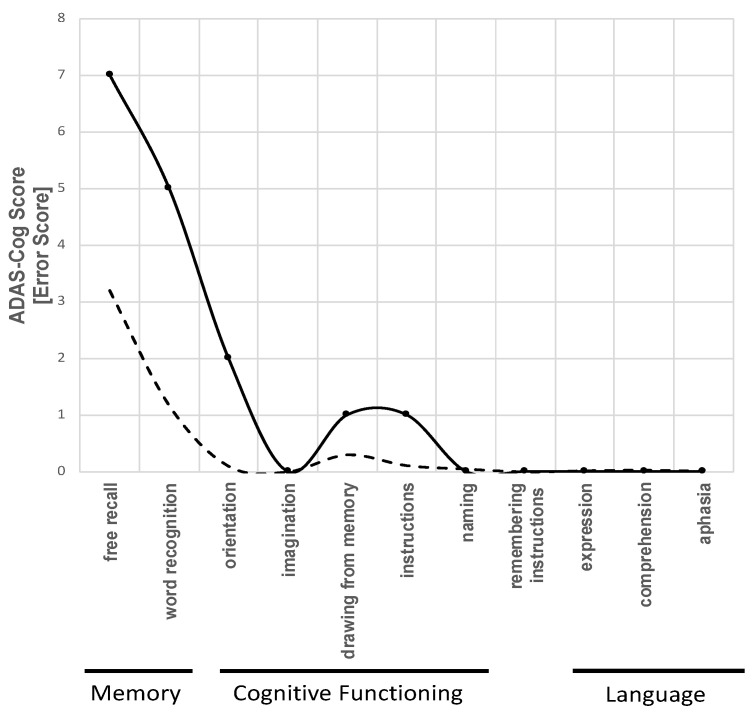
Example of a patient’s cognitive profile of the Alzheimer’s Disease Assessment Scale–Cognition (ADAS-Cog) with mild to moderate dementia from a patient included in the IMAD trial (own figure). Continuous line, cognitive profile with mild to moderate dementia; dashed line, cognitive profile with no cognitive impairment; grey area, 95% confidence interval for the cognitive profile. Without impairment according to Graham, Emery and Hodges [48].

**Figure 3 jcm-09-01919-f003:**
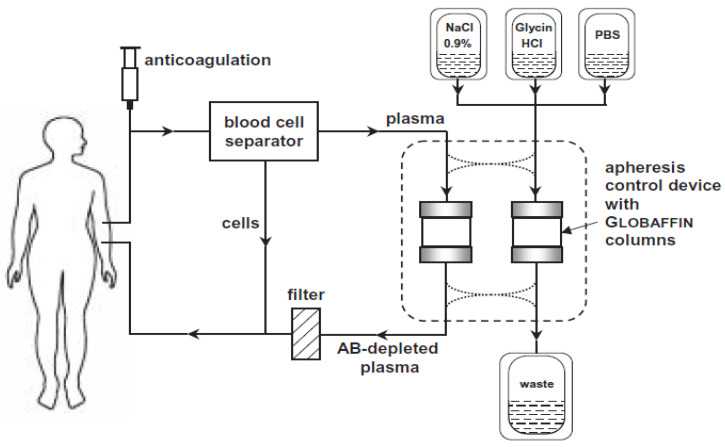
Principle of immunoadsorption. Principle of the immunoadsorption treatment. Reprinted from [68] with permission of John Wiley and Sons.

**Table 1 jcm-09-01919-t001:** Inclusion and exclusion criteria.

Criteria	
Inclusion	- Age 55–85
	- Previous or suspected diagnosis of Alzheimer’s clinical syndrome
	- Presence of agAAB against alpha1-adrenoceptor (α1AR)
	- Mini mental state examination score between 19 and 26
	- Written informed consent
Exclusion	- Presence of autoantibodies against the NMDA receptor
	- Defective blood coagulation at time of inclusion
	- Severe protein deficiency disorders
	- Known manifest vitamin/folic acid deficiency (substitution allowed)
	- Active infectious disease, or sings of ongoing infection with CRP >10 mmol/L
	- Impaired renal function (serum creatinine >220 µmol/L)
	- Any disease requiring immunosuppressive drugs or therapeutic antibodies
	- Non-curative treated malignant disease or another life-threatening disease with poor prognosis (estimated survival less than 2 years), except for basal-cell carcinoma
	- Unstable angina pectoris, second or third degree atrioventricular block or symptomatic sick sinus syndrome without implanted pacemaker, history of myocardial infarction, bypass or other revascularization procedures, valvular heart defect (≥2. degree)
	- Severely reduced left ventricular systolic function (LVEF <30%) and/or heart failure symptoms according to NYHA class III/IV
	- Clinical manifestation of arterial disease, vascular surgery: ACl-Stenosis >60%; PAD > IIb, history of stroke, diffusion disorder or expired territorial stroke in MRI
	- Endocrine disorder excluding diabetes mellitus
	- Severe hepatic disorder (Child–Pugh score 5 or more)
	- Drug therapy against dementia since less than 3 months
	- Psychopharmacological drug therapy since less than 3 months - Dialysis requirement
	- MRI contraindications (e.g., pacemaker) - Legal tutelage
	- Previous treatments with IA or immunoglobulins
	- ACE-treatment during the IA - Severe mental disorder (bipolar disorder, schizophrenia, depression) requiring treatment - Alcohol or drug abuse
	- Inability to undergo the study procedure - Participation in any other clinical/interventional study within less than 30 days prior to screening

ACE, angiotensin converting enzyme; ACI, internal carotid artery; agAAB, agonistic autoantibodies; CRP, C reactive protein; IA, immunoadsorption LVEF, left ventricular systolic function; MRI, magnetic resonance imaging; NMDA, N-methyl-D-aspartate; NYHA, New York Heart Association; PAD, peripheral artery disease.

**Table 2 jcm-09-01919-t002:** Methods, main outcome parameters and time points.

Method	Parameter	Screening	Baseline	Follow-Ups	Comments
Arterial Spin Labelling MRI	Cerebral blood flow		X	X	
MRI Basic protocol	Brain Volume, WMH, CBM; MTA		X	X	
Time-of-flight MR angiography	Vessel anatomy and size		X	X	
Diffusion Tensor Imaging	Fractional anisotropy		X	X	
ADAS-Cog	Cognition		X	X	
MMSE	Cognition	X	X	X	
GDS	Depression		X	X	
VLMT	Cognition		X	X	
Benton Test	Cognition		X	X	
Brachial blood	Brachial blood pressure values	X	X	X	
Pulse wave analysis	Arterial stiffness, central hemodynamics		X	X	
Digital endothelial vascular function and stiffness	Endothelial function and vascular stiffness		X		
Echocardiography	Cardiac function and structure		X		
Transcutaneous oxygen measurement	Oxygenation		X	X	
Kidney sonography	Renal function		X	X	
Doppler Sonography	Carotid Arteria blood flow		X	X	
Liquor analytics	Tau/P-Tau		(X)	(X)	Optional
Liquor analytics	ß-A40/42		(X)	(X)	Optional

IA, immunoadsorption; MRI, magnetic resonance imaging; CBM, cerebral microbleeds; MTA, medial temporal lobe atrophy; DTI, diffusion tensor imaging; ADAS, Alzheimer’s Disease Assessment Scale; MMSE, Mini-Mental State Examination; GDS, Geriatric Depression Scale; VLMT, Verbal Learning and Memory Test; LVEF, left ventricular ejection fraction; WMH, white matter hyperintensities; optional parameters are depending upon patient agreement; follow-up visits are conducted 1, 6 and 12 months after IA treatment.

**Table 3 jcm-09-01919-t003:** Assessments by standardized questionnaires.

Test	Abbreviation	No Impairment	Severe Impairment	Normal Range
Mini-Mental State Examination	MMSE	30	0	26–30
Alzheimer’s Disease Assessment Scale—cognitive Scale	ADAS-Cog	0	70	0–4
Memory		0	22	-
Cognitive Functioning		0	28	-
Language		0	15	-
Verbal Learning and Memory Test	VLMT			
Learning		75	0	48–75
Loss after Interference		0	15	-
Loss after Delay		0	15	0–3
Recognition		0	15	-
Benton Test	Benton Test			
Number Correct Score		10	0	-
Error Score		0	-	-
Geriatric Depression Scale	GDS	0	15	0–5

ADAS-Cog, Alzheimer’s Disease Assessment Scale–Cognition; VLMT, GDS, Geriatric Depression Scale; Verbal Learning and Memory Test.

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
