# Peer review of "Immunoadsorption for Treatment of Patients with Suspected Alzheimer Dementia and Agonistic Autoantibodies against Alpha1a-Adrenoceptor—Rationale and Design of the IMAD Pilot Study"

_jcm, 2020, doi:10.3390/jcm9061919_

Round 1

Reviewer 1 Report

This manuscript reports the protocol of a phase 2 study evaluating a new type of treatment - immunoadsorption of certain autoantibodies - in Alzheimer's disease.
This study is original and welcome because it explores a new pathophysiological hypothesis in a context where more conventional approaches have all failed. This is well explained in the introduction.
Main points:
1. The study should only include patients with probable Alzheimer's disease as defined by internationally validated criteria (DSM5 or other), and not patients with suspected clinical Alzheimer's syndrome as indicated in the inclusion criteria. The process and the criteria dor AD diagnosis should be explicited.
2. The main objective - the measurement of brain perfusion by MRI - is certainly questionable. Brain perfusion may vary from day to day or from time to time, independent of any treatment. The authors do not indicate what the primary endpoint will be (ratio of perfusion of certain regions of interest / overall brain perfusion?). It is mandatory to explicit the judgment criteria. The choice of a cognitive test would have been simpler, although due to the absence of a control group the interpretation is complex.
3. The indication of the GDS scale among the cognitive assessment in several tables is an error. As indicated in the text, it is a scale to look for symptoms of depression (not cognition).
4. As loss of independence is a major characteristic of patients with Alzheimer's disease, the authors should measure it at baseline and during follow-up. Similarly, there could be a search for psychological and behavioural symptoms of dementia as for example with the NPI scale.
5. Blind assessment. Since this is an open study, it would be prudent to adopt a blind assessment procedure, i.e., those measuring the endpoints (radiologists, neuropsychologists, or others) do not know the purpose of the study, or with a mixture of the order of radiological examinations for brain flow or other measurements. Otherwise there is a risk that the results may be influenced by subjective factors, whether or not the evaluators are aware of them.
6. Due to the large number of variables which will be tested, the p value should be adjusted according to Bonferroni method, and only comparisons of previously defined judgment criteria should be included in the analysis.

Minor points
The references concerning dementia in the world are quite old, there are more current ones.
The authors should indicate whether agonist autoantibodies are also present in elderly people without dementia, or with dementias other than Alzheimer's disease.
Do these autoantibodies exist in mouse models of Alzheimer's disease, and if so, has immunoadsorption been tried?

Reviewer 2 Report

Introduction: minor edit page 6

Methods: minor revisions.
Did the authors take genetic predisposition into account, specifically hetero.homozygous for apo-e4?
With that, the authors provide appropriate in/exclusion criterion, sampling and clinical/laboratory analyses.

Discussion:I would like to see strengths and limitations
The time period of one year should be discussed. I recognize that this is a study regarding justification for a pilot study, but, the follow up period makes associations and further causality difficult to establish.

The manuscript presented provides rationale for a novel approach to the on going issue regarding the lack of treatment modalities for Alzheimers dementia and disease. Utilization of immunoadsorption to slow the progression of dementia is truly out-of-the-box thinking. The authors have been quite thoughtful in their approach of this treatment modality. They have taken Alzheimer's and vascular type dementia into account. The information presented here provides adequate evidence to warrant a pilot study to establish an association between Alzheimer's clinical syndrome and removal of α1AR-AABs.

Round 2

Reviewer 1 Report

The authors took the reviewer's comments into account and substantially improved their manuscript and certain aspects of the study protocol. In particular, they clarified the main judgement criteria and planned a blind review process for it.

A major criticism regarding the patients participating in the study is not resolved by the authors in the revised version. The authors state in the title and in many parts of the manuscript that they are studying a treatment for "Alzheimer's dementia", but the patients in the study do not fit this diagnosis.

The authors argued that the patient should have a "previous or suspected diagnosis of clinical Alzheimer's syndrome". They do not explain how this inclusion criterion is verified or established. This diagnosis is not straightforward ("according to ref. 39, "clinical Alzheimer's syndrome - recommended terminology for clinically established multiple (or single) domain amnesic syndrome or a classic syndromic variant (i.e., what has historically been labeled "possible or probable AD")"). It applies to both mildly affected and demented individuals"). In my opinion, such a diagnosis requires a very detailed clinical and cognitive assessment. A single MMSE between the ages of 19 and 26 is not sufficient to define this condition. The "suspected diagnosis of clinical Alzheimer's syndrome" in non sense.

The authors partially agree with this point, but this cannot be changed since the study is ongoing. I think this major point is a serious flaw for the study described in the protocol. Possibly the authors could change the name defining the participants of their study as persons with cognitive imparment which better fit with the inclusion criteria related to MMSE score. Also, they should not claimed that they future findings applyes to AD patients.
